

# Botanical characteristics, chemical components, biological activity, and potential applications of mangosteen

Chenchen Bi[1,2,*], Hang Xu[1,*], Jingru Yu[1], Zhinan Ding[2] and Zheng Liu[2]

[1] Department of Clinical Medicine, School of Medicine, Shaoxing University, Shaoxing, Zhejiang, PR China
[2] Department of Pharmacology, School of Medicine, Shaoxing University, Shaoxing, Zhejiang, PR China
[*] These authors contributed equally to this work.

## ABSTRACT

*Garcinia mangostana L.* (Mangosteen), a functional food, belongs to the *Garcinaceae* family and has various pharmacological effects, including anti-oxidative, anti-inflammatory, anticancer, antidiabetic, and neuroprotective effects. Mangosteen has abundant chemical constituents with powerful pharmacological effects. After searching scientific literature databases, including PubMed, Science Direct, Research Gate, Web of Science, VIP, Wanfang, and CNKI, we summarized the traditional applications, botanical features, chemical composition, and pharmacological effects of mangosteen. Further, we revealed the mechanism by which it improves health and treats disease. These findings provide a theoretical basis for mangosteen's future clinical use and will aid doctors and researchers who investigate the biological activity and functions of food.

## INTRODUCTION

*Garcinia mangostana L.* (Mangosteen), also known as *Mangji persimmon*, belongs to the family *Garcinaceae* and is usually grow in Southeast Asian countries, such as Thailand, Malaysia, and Indonesia (*Aizat et al., 2019*; *Arif et al., 2020*). It is widely believed that this fruit tree that reaches a height of 6 to 25 m is native to Sunda and the Maluku islands of east Indonesia (Fig. 1). Mangosteen is suitable for planting in low-altitude tropical areas, and its growth is relatively slow. Its ripe fruits are spherical and measure 5–8 cm in diameter. The inedible peel constitutes more than 50% of the fresh weight and is often discarded as waste (*Chen et al., 2021*). However, the mangosteen peel has traditionally been used for its medical properties all over the world, especially in Southeast Asia (*Tousian Shandiz, Razavi & Hosseinzadeh, 2017*). Mangosteen has attracted attention both in its country of origin and Western countries, where it is regarded as a functional food. Its commercial products are supplied as extract powder, capsules, and juice (*Kim et al., 2015*). For example, it is becoming a household fruit juice in America due to its high concentration of antioxidant

Corresponding authors
Zhinan Ding,
dingzhinan1984@163.com
Zheng Liu, liuzheng1707@163.com

A

B

Figure 1 (A) An image of the unripe fruit on the mangosteen tree; (B) an image of the ripe fruit and its internal structure.

compounds, including phenolic compounds and $\alpha$-mangostin ( $\alpha$-MG) (*Kwansang, Chen & Chaiprateep, 2022*).

Current research indicates that mangosteen contains abundant chemical constituents, including polysaccharides, xanthrones, procyanidin, benzophenones, bioflavonoids, and triterpenoids (*Espirito Santo et al., 2020*). The pharmacological effects of mangosteen include anti-oxidative, anti-inflammatory, anti-tumor, anti-depressive, anti-microbial anti-parasitic, and neuroprotective effects (*Markowicz et al., 2019*; *Sultan et al., 2022*). It has also been used in the treatment of various skin conditions. However, to the best of our knowledge, the traditional applications, botany, chemical composition, and pharmacological effects of mangosteen, and especially the mechanism by which it improves health and treats disease, have not been reviewed comprehensively. Therefore, in this review, we summarized and analyzed the data presented in recent studies on mangosteen with the goal of expanding the collective knowledge on mangosteen and its medicinal applications.

## SURVEY METHODOLOGY

PubMed, Science Direct, Research Gate, Web of Science, VIP, Wanfang, CNKI, and other databases were used to search for the following terms in English or Chinese: mangosteen, botanical features, phytochemistry, biological activity, and application.

## TRADITIONAL APPLICATIONS

Mangosteen is a medical plant with a long history in Southeast Asia, including Malaysia and the Philippines (*Wang et al., 2018*). In the ayurvedic system of medicines which is widely practiced in modern India, mangosteen has high medicinal value because its hull contains $\alpha$-MG, which has been used for the treatment of inflammation, diarrhea, cholera, and dysentery (*Ibrahim et al., 2019*). In addition to the hull, the rind, bark, and root have also been shown to have medicinal value. The rind has been utilized as an

antibacterial agent and a medicine for the management of wounds, suppurations, and chronic ulcers. Hyperkeratosis, eczema, psoriasis, and other skin disorders have been treated by an ointment extracted from the leaves and bark of mangosteen (*Gopalakrishnan, Banumathi & Suresh, 1997*). In addition, amibiasine is a bark extract employed for the therapy of amoebic dysentery. The rind has been used for the treatment of gleet, cystitis, gonorrhea, and diarrhea (*Taher et al., 2016*). A root decoction has been utilized to manage menstrual disorders in women. A decoction and infusion extracted from the peels and seeds has shown anti-pyretic, anti-scorbutic, anti-infective, and laxative effects. Specifically, it has been prescribed for gastrointestinal and urinary tract infections. The rinds and decoctions of barks and leaves are the raw materials of a tea, which has been used to treat different urinary disorders, dysentery, diarrhea, and fever in Malaya and the Philippines (*Obolskiy et al., 2009*). In conclusion, although different parts of mangosteen have their own therapeutic effects on diseases, the relevant active constituents and mechanisms of action have not yet been elucidated. Therefore, studying the extraction, isolation, identification, pharmacological actions, and underlying mechanisms of active parts of mangosteen is of great significance.

## BOTANY

Mangosteen belongs to the *Garcinaceae* family and is an endemic evergreen tree species that is suitable for planting in low-altitude areas (*Matra et al., 2016*). It is a strictly tropical fruit with a restricted range of adaptability. Thus, it can only be found in Malaysia, Indonesia, Thailand, southern India, Northern Australia, Hawaii, Central America, Brazil, and other tropical countries. Although mangosteen is a complex dioecious plant, males are not yet known, and females have sterile anthers, resulting in anathecus reproduction (*Ramage et al., 2004*). Therefore, due to its strict growth environment requirements and difficult breeding conditions, mangosteen is a precious fruit.

The mangosteen plant can reach a height of 6–25 m, with dense foliage covering the crown. The growth of mangosteen is relatively slow, as it takes 7–9 years for the first harvest after planting. The fruit matures from May to October every year, and the yield is highest from August to October (*Aizat et al., 2019*). After ripening, the fruit is spherical, dark purple, and 5–8 cm in diameter. It has white flesh and a hard rind. Due to its unique taste and medicinal value, mangosteen is known as the "Queen of Fruits." (*Ramage et al., 2004*). The edible aril is juicy and soft, with a sweet and slightly sour taste and a pleasant fragrance (*Khaw, Chong & Murugaiyah, 2020*). Moreover, mangosteen seeds are present in one or both septa of each fruit. Mangosteen is classified as a fragile plant due to its sensitivity to drought and low temperatures (*Eukun Sage et al., 2018*).

## PHYTOCHEMISTRY

### Polysaccharides

Mangosteen polysaccharides have a complex molecular structure, consisting of a variety of monosaccharides. They are extensively used in the biomedical field due to their non-toxicity and lack of side effects. These polysaccharides have antioxidant, anticoagulant, anti-tumor,

and immune system-supporting effects (*Chen et al., 2020*; *Park & Shin, 2019*; *Zhang et al., 2020a*; *Zhang et al., 2020b*). They also have unique structural features, such as the presence of specific functional groups, different degrees of branching, and unique conformations and molecular weight distribution. As a main bioactive component in mangosteen peel, polysaccharides are characterized by abundant and powerful bioactivities (*Zhang et al., 2020a*; *Zhang et al., 2020b*). The biological activities of polysaccharides depend on their structural features; thus, polysaccharides from the same source exhibit different biological activities.

The polysaccharide content of a mangosteen peel extract was 27.12%. In a study, researchers isolated and identified two kinds of polysaccharides from mangosteen peel, GMP70-1 and GMP90-1, and determined their structure and molecular characteristics. GMP70-1 consisted of four monosaccharides, namely, arabinose, arabinose, rhamnose, and galacturonic acid, with a molar ratio of 16.20:3.29:2.95:1.00, respectively. The repeated unit of GMP70-1 was determined to be $(1 \rightarrow 5)$-linked $\alpha$-L-Araf, $(1 \rightarrow 3, 5)$-linked $\alpha$-L-Araf, $(1 \rightarrow 2, 4)$-linked $\alpha$-L-Rhap, $(1 \rightarrow 4)$-linked $\beta$-D-Galp, terminating in t-$\alpha$-L-Araf, t-$\alpha$-D-GalpA, and t-$\beta$-D-Galp (*Zhang et al., 2020a*; *Zhang et al., 2020b*). The major components of GMP90-1 were arabinose, galactose, and rhamnose, with a relative abundance of 92.4%, 5.1%, and 2.6%, respectively (Fig. 2). The backbone of GMP90-1 was composed of $(1 \rightarrow 5)$-linked $\alpha$-L-Araf, $(1 \rightarrow 2,3,5)$-linked $\alpha$-L-Araf, $(1 \rightarrow 3,5)$-linked $\alpha$-L-Araf, $(1 \rightarrow 6)$-linked $\beta$-D-Galp, and $(1 \rightarrow 2)$-linked $\alpha$-L-Rhap. The two polysaccharides extracted from mangosteen promoted macrophage phagocytosis to neutral red; accelerated the release of tumor necrosis factor-$\alpha$ (TNF-$\alpha$), interleukin-1 $\beta$ (IL-1 $\beta$), interleukin-6 (IL-6), reactive oxygen species (ROS), and nitric oxide (NO); and showed significant immunomodulatory effects.

These results suggest that the anti-inflammatory effects of polysaccharides in mangosteen peel may be achieved through the regulation of inflammatory cytokines and antioxidant effects.

Plant polysaccharides have become popular research topics in functional chemistry and modern medicine (*Lu et al., 2021*). Further, polysaccharides from *Lentinula edodes* have immunomodulatory properties (*Roszczyk et al., 2022*), whereas those from *Ginseng polysaccharides* alter the gut microbiota, potentiating the antitumor effect of anti-programmed cell death 1/programmed cell death ligand 1 immunotherapy (*Huang et al., 2022*). The development and utilization of plant polysaccharides show great potential in the development of functional foods and medicine (*Tamang et al., 2022*). Currently, research on mangosteen polysaccharides is limited, and there have only been studies on their extraction, preliminary separation, identification, and basic functions (*Chitchumroonchokchai et al., 2013*; *Wathoni et al., 2019*; *Zou et al., 2021*). In the future, research should elucidate the *in vivo* and *in vitro* pharmacological effects of these polysaccharides and their mechanisms. It is expected that mangosteen series products can exert superior health effects in comparison to other foods and medical products.

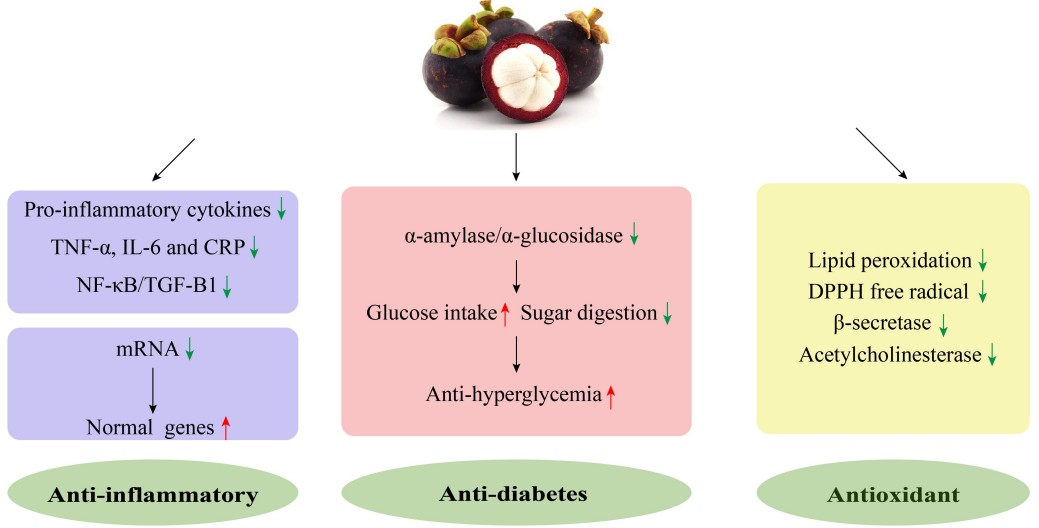

**Figure 2** **The mechanism of the anti-inflammatory, antioxidant, and anti-diabetes properties of mangosteen.** Mangosteen could exert anti-inflammatory activity *via* downregulating the expression of pro-inflammatory cytokines and transcription factors NF-$\kappa$B/TGF-B1. (A) Reduction in TNF-$\alpha$, IL-6, and CRP and a decrease in mRNA levels could restore the expression of these genes to normal levels. Mangosteen could exert anti-hyperglycemia activity against diabetes by inhibiting $\alpha$-amylase/ $\alpha$-glucosidase, promoting glucose intake, and reducing sugar digestion. Mangosteen could exert antioxidant activity by reducing lipid peroxidation, eliminating DPPH free radical, and inhibiting the activities of $\beta$-secretase and acetylcholinesterase.CRP, C-reactive protein; IL-6, interleukin-6; NF-$\kappa$(B), nuclear factor $\kappa$-B; TGF-B1, transforming growth factor-B1; TNF-$\alpha$, tumor necrosis factor-$\alpha$.

## Xanthone

Xanthone is the most prevalent phytochemical component in mangosteen and may contribute to its high quality (*Ryu et al., 2010*). The most representative xanthone isolated from mangosteen peel is $\alpha$-MG, which has a wide range of biological activities and pharmacological effects. The pharmacological effects of $\alpha$-MG include antioxidant, anticancer, anti-obesity, antibacterial, and antimalarial properties, in addition to neuroprotective, liver-protective, and heart-protective properties in Alzheimer's disease (AD) (*Abate et al., 2022*; *Eisvand et al., 2022*; *Labban et al., 2021*; *Meepagala & Schrader, 2018*; *Oh et al., 2021*; *Wang et al., 2021*; *Zhu et al., 2021*). Because of their poor water solubility, $\alpha$-MG preparations typically require a high concentration of a solubilizer, limiting their use in certain clinical applications. Therefore, a new, local, $\alpha$-MG-containing nano-emulsion with an optimal oil phase and surfactant content has been developed. $\alpha$-MG can be loaded into the nano-emulsion without significant changes to its chemical and physical properties, and its maximum concentration was found to be 0.2% (w/w) (*Asasutjarit et al., 2019*). Although $\alpha$-MG has a variety of biological activities, the requirements for the effective use of $\alpha$-MG remain unclear, and further research is needed.

$\gamma$-Mangostin ($\gamma$-MG) is another important xanthone, with hydroxyl groups isolated from mangosteen peel through a microwave-assisted extraction and high-speed, counter-current chromatography (*Fang et al., 2011*). It has anti-cancer, anti-hyperglycemic, and

anti-malarial effects. $\gamma$-MG provides prolonged free compound exposure and subsequently has high *in vivo* activities. Other components could influence the metabolism of $\gamma$-MG by inhibiting the initial combination and increasing the content of free $\gamma$-MG. Only free $\gamma$-MG has beneficial biological activities (*Li et al., 2013*). These findings suggest that food supplements prepared from mangosteen extract are superior to pure compound products. However, further investigations are required to directly compare the *in vivo* activities of pure compounds and extracts in order to draw a final conclusion.

## Procyanidin

Procyanidin is a phenolic compound consisting of epicatechin and catechin (*Zheng et al., 2021*). The properties of procyanidins, which are water-soluble compounds in mangosteen peel, include anti-inflammatory and antibacterial effects (*Oh et al., 2021*). The main polyphenols of mangosteen peel are epicatechin and proanthocyanidin B2 (*Wang, Chen & Wang, 2017*). Epicatechin is a flavanol compound with a molecular formula of $C_{15}H_{14}O_6$. It has free radical-scavenging, antioxidant, and skin-protective properties (*Ngawhirunpat et al., 2010*). Proanthocyanidin B2 is an epicatechin dimer with antioxidant, anti-inflammatory, and anti-carcinogenic activities (*Liu et al., 2020*; *Zuriarrain et al., 2015*). Uniquely, proanthocyanidin B2 also has a lipopolysaccharide (LPS)-binding capacity, which enables its accurate identification (*Zheng et al., 2021*).

## Benzophenones

Benzophenone synthase (BPS) belongs to the type III polyketide synthase superfamily, and it is an essential enzyme in the biosynthesis of xanthone (*Songsiriritthigul et al., 2020*). It also catalyzes the biosynthesis of benzophenone (*Nualkaew et al., 2012*). Therefore, benzophenones are common bioactivators in mangosteen. Many benzophenones with polyisoprenylated benzophenone skeletons, which could be extracted from the *Garcinaceae* family, have been reported (*Choodej et al., 2022*). Benzophenones have antioxidant, anti-inflammatory, anti-cancer, cytotoxic, and anti-bacterial activities. Benzophenones exhibit a great structural diversity, having more than 3,000 members. Among of them, 2,3′,4,6-tetrahydroxybenzophenone, 3,4,5,3′-tetrahydroxybenzophenone, 2,4,6,3′,5′-pentahydroxybenzophenone, 1,3,6,7-tetrahydroxyxanthone, 2,3′,4,4′-tetrahydroxy-6-methoxybenzophenone, congestiflorone, 2,3′,4,5′-tetrahydroxy-6-methoxybenzophenone, mangaphenone, and benthamianone have been extracted from mangosteen (*Jiang et al., 2010*; *See et al., 2021*).

## Phenolic acid

*Zadernowski, Czaplicki & Naczk (2009)* identified 10 phenolic acids from a mangosteen fruit extract. Phenolic acids, which are aromatic secondary plant metabolites, account for about one-third of the total phenol content in mangosteen fruits. They are widely found in plants in the free and bound forms (*Robbins, 2003*). Bound phenolics could bind to a variety of plant components through acetal, ester, or ether bonds (*Chalas et al., 2001*). In mangosteen, the major phenolic acids in the aril, peel, and rind are p-hydroxybenzoic acid and protocatechuic acid. m-Hydroxybenzoic acid was only found in the peel, while

3,4-dihydroxymandelic was only present in the rind. Phenolic acids have antimicrobial and antioxidant properties, and they bind specifically to human serum albumin (Table 1).

## PHARMACOLOGY

### Antioxidant properties

The incidence of degenerative diseases can be reduced *via* the incorporation of fruits and vegetables in the diet (*Abate et al., 2022*; *Zhang et al., 2020a*; *Zhang et al., 2020b*). Various antioxidants found in fruits and vegetables, such as tea polyphenols and vitamin E, are regarded as their key active compounds (*Chandran & Abrahamse, 2020*; *Petruk et al., 2018*; *Urquiaga & Leighton, 2000*). Antioxidants are known to mitigate harmful oxidative reactions. Many studies have shown that lipids, proteins, and nucleic acids undergo oxidative damage caused by free radicals (*Di Meo & Venditti, 2020*; *Pisoschi & Pop, 2015*). Oxidative stress induces lipid peroxidation, which is the key process of many pathological events (*Guéraud et al., 2010*). Malondialdehyde (MDA), the decomposition product of unsaturated lipid oxidation, destroys membrane lipids and is known to have mutagenic and carcinogenic effects (*Ayala, Muñoz & Argüelles, 2014*). Antioxidants are very important in preventing these diseases, as the production of oxidizable substrates could be inhibited or delayed by the activity of antioxidant compounds.

*Tjahjani (2017)* found that various mangosteen rind (GMR) fractions had a high 1,1-diphenyl-2-picrylhydrazyl (DPPH) capture activity, indicating mangosteen's good antioxidant activity. *Hassan et al. (2021)* found that GMR could significantly reduce the redox imbalance and toxicity of liver tissue caused by a long-term $\gamma$-ray irradiation. It also improved the function, MDA content, antioxidant enzyme activity, and NO level of damaged liver. *Oh et al. (2021)* demonstrated that an aqueous extract of mangosteen peel powder reduced lipid peroxidation and eliminated DPPH free radicals in cultures of primary rat cortical cells, confirming its antioxidant properties. The extract also inhibited the activities of $\beta$-secretase and acetylcholinesterase, leading to antioxidant and neuroprotective effects. Antioxidants are very important in preventing these diseases, as the production of oxidizable substrates could be inhibited or delayed by the activity of antioxidant compounds (*Chang et al., 2020*).

### Anti-inflammatory properties

Inflammation is a natural defense response of an organism to a stimulation involving pathogens, toxic agents, or a tissue injury. Many inflammatory cells and mediators participate in inflammatory responses. In most cases, the inflammatory response plays an advantageous role for organisms, but sometimes it leads to cellular damage. Many vegetables and fruits, including mangosteen, have anti-inflammatory effects (*Li et al., 2018*; *Tatiya-aphiradee, Chatuphonprasert & Jarukamjorn, 2021*). *Park et al. (2021)* showed that mangosteen pericarp extract (GME) decreased mRNA levels, restored the expression of genes to normal levels, and downregulated the expression of pro-inflammatory cytokines. These changes supported the therapeutic action of GME on an MRSA-induced superficial skin infection in mice. *Hassan et al. (2021)* found that GMR significantly reduced inflammatory markers, such as TNF-$\alpha$, IL-6, and C reactive protein (CRP), and

Peer J

**Table 1  The chemical constituents and function of mangosteen.**

| | Bioactive Compounds | Function | Reference |
|---|---|---|---|
| Polysaccharide | GMP70-1<br>GMP90-1 | Promote the phagocytosis of neutral red by macrophages and the secretion of NO, ROS, tumor necrosis factor-$\alpha$,IL-6 and IL-1 $\beta$, and have significant immunomodulatory effects | *Zhang et al. (2020b)*<br>*Zhang et al. (2020a)* |
| Xanthrone | $\alpha$-MG | Anticancer, antioxidant, antibacterial, antimalarial and anti-obesity activities, as well as neuroprotective, liver protective and heart protective properties in Alzheimer's disease (AD) | *Ryu et al. (2010)* |
| | $\gamma$-MG | Anticancer, anti-hyperglycemia, antimalarial effects | |
| Procyanidin | Epicatechin | Antioxidant, free-radical scavenging, and skin protective activity | *Yoshimura et al. (2015)* |
| | Proanthocyanidin B2 | Antioxidant activity, anti-inflammatory efficacy and anti-carcinogenic properties | |
| Benzophenones | 2, 3′, 4, 6-tetrahydroxybenzophenone<br>3, 4, 5, 3′-tetrahydroxybenzophenone<br>2,4,6,3′, 5′-pentahydroxybenzophenone<br>1,3,6,7-tetrahydroxyxanthone<br>2,3′, 4, 4′-tetrahydroxy-6-methoxybenzophenone<br>2,3′, 4, 5′-tetrahydroxy-6-methoxybenzophenone<br>Congestiflorone<br>Mangaphenone<br>Benthamianone | Cytotoxicity, antioxidant activity, anti-inflammatory activity, anticancer activity, and antibacterial activity | *Choodej et al. (2022)*<br>*See et al. (2021)*<br>*Jiang et al. (2010)* |
| Phenolic acid | Protocatechuic acid<br>P-hydroxybenzoic acid<br>M-Hydroxybenzoic acid<br>3,4-dihydroxymandelic | Antioxidant activity | *Zadernowski, Czaplicki & Naczk (2009)* |

downregulated the transcription factors NF-$\kappa$B/TGF-B1 in liver tissue after long-term $\gamma$-ray irradiation. At the same time, GME has also shown good therapeutic effects on gingivitis and early periodontitis. Moreover, an oral mangosteen and propolis extracted complex has the clinical and immunological potential to reduce gingivitis (*Zhang et al., 2022*). These are important examples of the anti-inflammatory effects of mangosteen, and further studies should explore the specific bioactive compounds and mechanisms involved (Fig. 2).

## Anti-tumor properties

Cancer is a great threat to human health and life. Currently, chemical therapy is still one of the most important cancer treatment methods. Molecules of unmodified natural products, their semi-synthetic derivatives, or synthetic biosimilars account for about 50% of the more than 200 new chemical entities approved for anti-cancer use in the past 50 years (*Butler, Robertson & Cooper, 2014*; *Newman & Cragg, 2016*). Several natural, plant-derived products have demonstrated good anti-tumor properties (*Zou et al., 2021*). According to current studies, mangosteen and its bioactivators have potential applications in cancer therapy (*Agarwal et al., 2020*; *Taokaew et al., 2021*).

*Zhu et al. (2021)* found that the migration and invasion of breast cancer cells was inhibited by $\alpha$-MG due to the cleavage of poly ADP-ribose polymerase (PARP) and apoptosis *via* the PI3K/protein kinase B (AKT) signaling pathway targeting RXR $\alpha$. In addition, the proliferation of multiple breast cancer cells was inhibited by $\alpha$-MG *via* a reduction in cancer compounds (*Herdiana et al., 2021*). Fatty acid synthase (FAS) is a target of breast cancer treatment due to its overexpression in human breast cancer cells. *Li, Tian & Ma (2014)* demonstrated that $\alpha$-MG decreased the accumulation of intracellular fatty acids *via* the inhibition of the intracellular activity and expression of FAS.

$\alpha$-MG has also exhibited anti-cancer and anti-proliferative properties in various digestive cancers, such as advanced hepatocellular carcinoma (HCC) and colorectal cancer (CRC) (*Chitchumroonchokchai et al., 2013*). *Wang et al. (2021)* found that the combination of $\alpha$-MG and sorafenib could inhibit the proliferation of HCC cells and induce apoptosis. Moreover, the authors demonstrated the safety and effectiveness of this combined therapy in a mouse model. *Mohamed et al. (2017)* extracted 14 new mangosteen peel compounds and found that carnation flavone E showed considerable anti-proliferative effects/cytotoxicity, which induced a significant cell cycle stagnation in the G0/G1 phase. Ultimately, it demonstrated a cytotoxic effect, leading to the necrosis and apoptosis of human CRC and HCC cells. However, mangostanone IV (MX-IV) only induced necrosis and apoptosis in CRC cells. Further, Wu et al. showed that $\gamma$-MG downregulated GSK3 $\beta$-related signals. In addition, it overcame 5-fluorouracil-induced resistance in cancer-related fibroblasts and hindered the production of cancer stem cells in CRC (*Wu et al., 2022*).

The potential of mangosteen-derived compounds as emerging cancer therapies is worthy of further study. Bioactive substances isolated from mangosteen fruits have shown their effectiveness as natural cytotoxic agents against certain cancers (*Akao, Nakagawa & Nozawa, 2008*; *Li et al., 2017*). These studies provide important scientific evidence for the potential application of mangosteen in cancer treatment. In fact, compared with Cancer

medicines available in the present-day pharmaceutical industries such as Methotrexate, Fluorouracil and Alkeran, potential natural candidates with anti-cancer properties (flavonoids, ganoderma lucidum polysaccharide, lentinus edodes polysaccharide, *etc.*) have fewer adverse reactions, and originated from natural food or Chinese herbal medicine, which are easy to obtain, economic, easy to be accepted by patients (*Uddin et al., 2020*; *Man et al., 2012*).

## Diabetes prevention and treatment

Diabetes mellitus (DM) is marked by high blood glucose resulting from the body's insensitivity to the actions of insulin (*Dennery, 2006*). *Chen et al. (2021)* found that $\gamma$-MG inhibited $\alpha$-amylase/ $\alpha$-glucosidase through insulin sensitization, promoted glucose intake, and reduced sugar digestion, thus exerting an anti-hyperglycemic activity. Protein tyrosine phosphatase 1B (PTP1B) is an action site for diabetes medicines. *Hu et al. (2021)* reported that garcinone E, a xanthone of mangosteen, effectively inhibited the activity of PTP1B. Moreover, *Watanabe et al. (2018)* found that a mangosteen extract significantly improved the insulin sensitivity of insulin-resistant obese female patients without side effects. In conclusion, mangosteen has multiple bioactivities toward DM, providing new strategies for the discovery and development of diabetes drugs. However, the therapeutic effect of mangosteen on type 1 diabetes has not been thoroughly studied.

## Therapy for central nervous system (CNS) diseases

CNS diseases include Parkinson's disease (PD), AD, depression, stroke, and sleep disorders (*Sharma et al., 2019*). The major pathologies of these diseases involve oxidative stress, neuroinflammation, reduction of neurotrophic factors, and mitochondrial dysfunction (*Do & Cho, 2020*). Recently, many researchers reported the anti-inflammatory, anti-oxidative, and neuroprotective effects of mangosteen (*Oh et al., 2021*; *Weecharangsan et al., 2006*). Herein, we summarize its bioactivities and clinical application in these diseases.

### *Alzheimer's disease*

AD is a progressive neurodegenerative disease that is characterized by hyperphosphorylation of the tau protein, inducing the deposition of amyloid and neurofibrillary tangles. Abdallah et al. discovered that MX-IV reduced oxidative stress, neuroinflammation, and apoptosis. In addition, through the regulation of the PI3K/Akt/GSK-3 $\beta$ pathway, the content of nicotinamide adenine dinucleotide phosphate (NADPH) oxidase, activity of cleaved caspase-3, number of amyloid plaques, and expression of phosphorylated tau decreased. These changes led to a significant enhancement of neuronal survival and cognitive abilities (*Abdallah et al., 2021*). Ruankham et al. reported that $\alpha$-MG inhibited the activation of caspase-3/7, reduced the BAX protein, and increased the anti-apoptotic BCL-2 protein. In turn, these effects inhibited cell death caused by oxidative stress in neurons. At the same time, $\alpha$-MG directly binds to the active site of SIRT1, activating the SIRT1/3-FOXO3a pathway. Thus, it is a promising anti-oxidative stress therapeutic compound for the treatment of AD (*Ruankham et al., 2022*). These studies demonstrate the potential of mangosteen as a functional food for the treatment of neurodegenerative diseases due to the high application value of its active ingredients. The mechanism implicated in

mangosteen's effects on AD and the efficacy of certain bioactive compounds require further investigation.

### Parkinson's disease

PD is characterized by "Lewy bodies," and it is the most commonly diagnosed movement disorder. Moreover, it is the second most prevalent neurodegenerative disease (*Raza, Anjum & Shakeel, 2019*). *Huang et al. (2020)* discovered that tovophyllin A (TA), a flavonoid extracted from mangosteen, reduced apoptosis in primary cortical neurons in PD. In an *in vivo* acute PD model, TA also alleviated behavioral dysfunctions and the loss of dopaminergic neurons. Therefore, the results illustrate that TA is a strong cytoprotective factor for dopaminergic neurons in a PD model. The study by *Hu et al. (2016)* revealed that $\alpha$-MG protected $\alpha$-synuclein-induced microglial cells from direct neurotoxicity *via* suppressing NF-kB and NADPH oxidase. Therefore, bioactive substances isolated from mangosteen fruits have shown powerful neuronal protective effects, making them therapeutic candidates for PD treatment.

### Neurological conditions

Monoamine dysregulation is the primary pathology of depression. *Oberholzer et al. (2018)* showed that a raw mangosteen extract, containing $\alpha$-MG and $\gamma$-MG, presented antidepressant-like effects. *Ghasemzadeh Rahbardar, Razavi & Hosseinzadeh (2020)* also found $\alpha$-MG to have therapeutic effects on neuropathic pain caused by a chronic compression injury due to its antioxidant, anti-inflammatory, and antiapoptotic properties.

## Anti-parasitic properties

Parasitic diseases prevail widely around the world, especially in developing countries located in the subtropical and tropical regions (*Momčilović et al., 2019*). In 2000, the World Health Organization listed 11 parasitic infectious diseases as neglected tropical diseases, as they threaten the health of millions of people and disproportionately affect the poor. Recent studies have demonstrated the potential of mangosteen in the prevention and treatment of parasitic diseases.

Malaria remains an important public health issue worldwide (*Brock et al., 2015*). Quinone reductase 2 (QR-2), a cytoplasmic enzyme in human erythrocytes, has been related to the pathogenesis of malaria and other diseases. Another study reported that $\gamma$-MG has a strong inhibitory effect on QR-2. Its interaction with QR-2 includes hydrogen bonding and aromatic-aromatic interactions. In a study by *Tjahjani (2017)*, GMR exhibited an antimalarial activity that was similar and synergistic to artemisinin. *Upegui et al. (2015)* showed that the activity of $\alpha$-MG against chloroquine-resistant strains of *Plasmodium falciparum* was higher than that of $\delta$-MG. In summary, mangosteen extract has a good antimalarial activity and could be used in the treatment of parasitic diseases.

In humans, *Acanthamoeba* spp. is the main pathogen implicated in acanthamoeba keratitis (AK) and granulomatous amoebic encephalitis (*Loufouma Mbouaka et al., 2021*). AK is a serious corneal infection that can sometimes lead to blindness if not diagnosed and treated in a timely manner (*de Lacerda & Lira, 2021*). In recent years, the number of patients with AK caused by *Acanthamoeba* spp. has increased significantly worldwide,

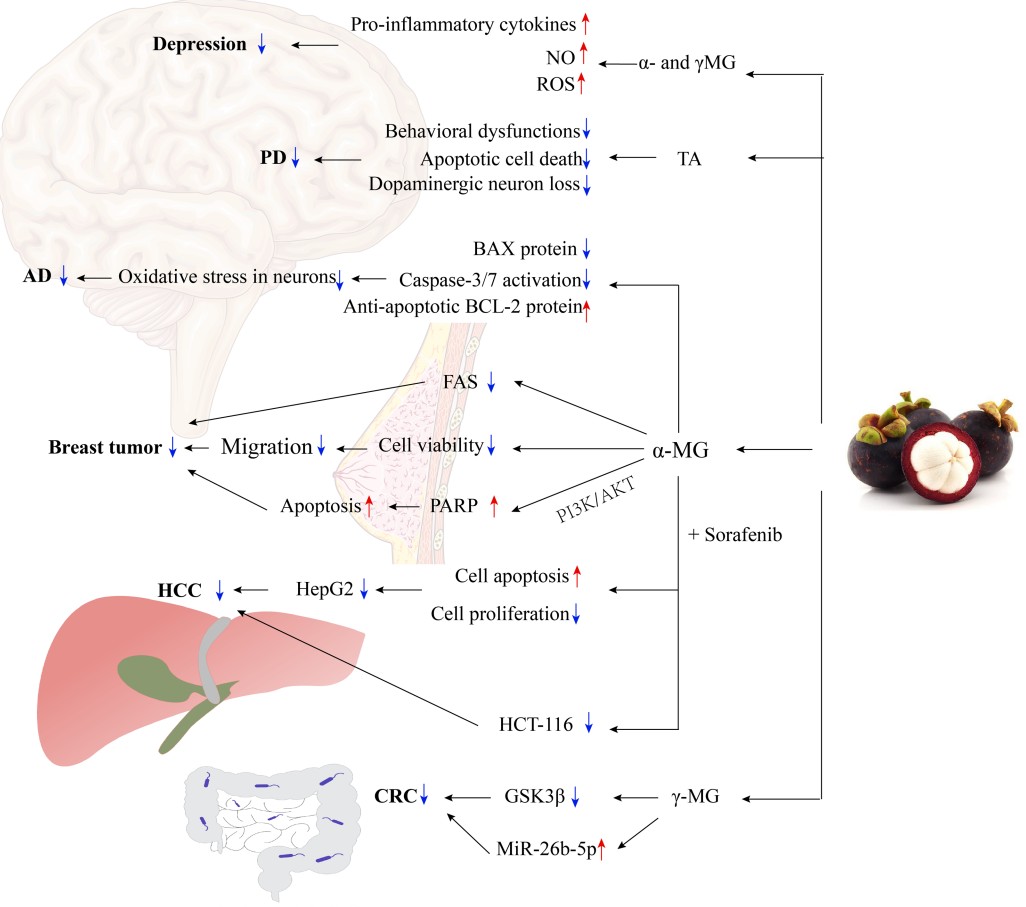

**Figure 3** **The molecular mechanisms of mangosteen in the treatment of CNS diseases and cancer.** TA could treat PD through the cytoprotective effect for dopaminergic neurons. $\alpha$-MG could treat AD by inhibiting cell death induced by oxidative stress in neurons. $\alpha$-MG could treat breast tumors by inhibiting the expression and intracellular activity of FAS, inhibiting the migration of breast cancer cells, and inducing apoptosis. $\alpha$-MG combined with sorafenib could treat HCC by enhancing cell apoptosis and inhibiting cell proliferation. $\gamma$-MG could treat CRC by inhibiting the GSK3 $\beta$-related signal pathway and increasing the level of miR-26b-5p.AD, Alzheimer's disease; $\alpha$-MG, $\alpha$-mangostin; CRC, hepatocellular carcinoma; FAS, fatty acid synthase; $\gamma$-MG, $\gamma$-mangostin; GSK3 $\beta$, glycogen synthase kinase 3 $\beta$ (GSK3 $\beta$); HCC, hepatocellular carcinoma; NO, nitric oxide; PARP, poly ADP-ribose polymerase; PD, Parkinson's disease; ROS, reactive oxygen species; TA, tovophyllin A.

and the infection has been associated to the increase in contact lens users (*Sifaoui et al., 2017*). *Sangkanu et al. (2022)* found that a mangosteen extract and $\alpha$-MG showed an anti-*Acanthamoeba* activity to trypanosome trophozoites and cysts and inhibited the growth of amoebas (*Tatiya-aphiradee, Chatuphonprasert & Jarukamjorn, 2019*). Furthermore, $\alpha$-MG eliminated the possibility of triangular adhesions from contact lenses. Further, (*Sangkanu et al., 2022*) demonstrated that crude mangosteen extracts and $\alpha$-Mg could be incorporated into contact lens solutions to effectively eliminate triangular adhesions within 24 h. Thus, mangosteen extracts and $\alpha$-MG are promising candidates for the treatment of *Acanthamoeba* infections (Fig. 3).

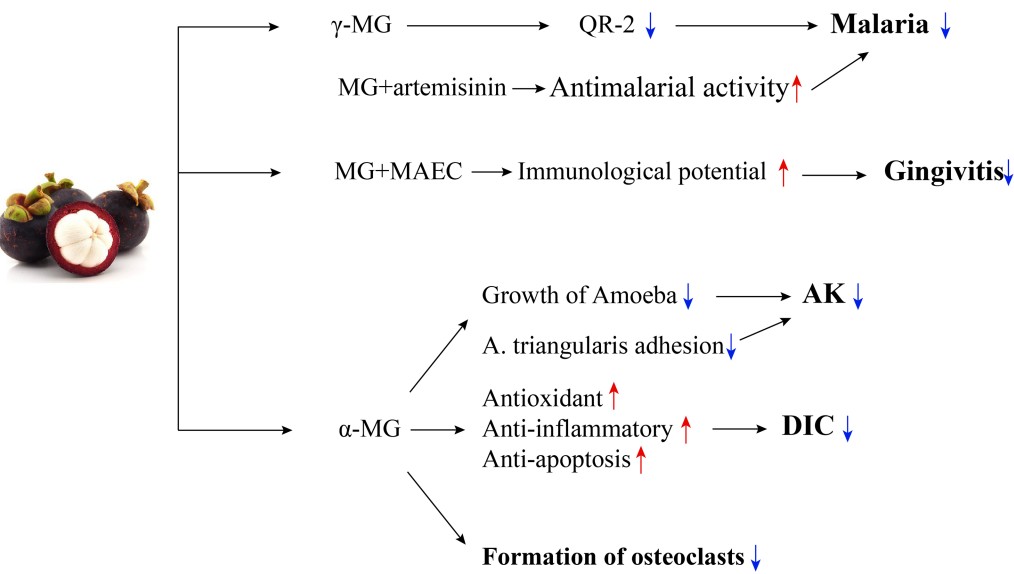

**Figure 4** **The mechanisms of mangosteen in the treatment of parasitic and other diseases.** $\gamma$-MG could manage malaria by inhibiting QR-2. MG has synergistic antimalarial activity with artemisinin. MG combined with MAEC could treat gingivitis, due to its immunological potential. $\alpha$-MG could treat AK by inhibiting the growth of amoebas and removing triangularis adhesion. $\alpha$-MG could treat DIC through its antioxidant, anti-inflammatory, and anti-apoptosis effects. $\alpha$-MG could inhibit the formation of osteoclasts.AK, acanthamoeba keratitis; $\alpha$-MG, $\alpha$-mangostin; DIC, doxorubicin-induced cardiotoxicity; $\gamma$-MG, $\gamma$-mangostin; MAEC, mangosteen and propolis extracts; MG, mangostin; QR-2, quinone reductase 2.

## Additional biological functions

In addition to the properties discussed above, mangosteen has an immunological activity, cardiovascular protective effects, and therapeutic effects in the treatment of gingivitis and early periodontitis. $\alpha$-MG has been shown to improve the cardiotoxicity of doxorubicin in chemotherapy without decreasing its anticancer effect (*Eisvand et al., 2022*). Also, researchers found that $\alpha$-MG can inhibit the formation of osteoclasts (*Zhang et al., 2022*). In summary, mangosteen is a valuable functional food with the potential to treat many systemic diseases. Thus, the extraction and effective utilization of the bioactive components from mangosteen peel need further investigation (Fig. 4; Table 2).

## TOXICOLOGY

Mangosteen contains a high proportion of polyphenols, polysaccharides, and other active substances. As a result, its antioxidant and probiotic properties, as well as its role in immunomodulation and hypoglycemic maintenance, have been widely investigated by researchers all over the world (*Kudiganti et al., 2016*). The mangosteen fruit has a high nutritional value, and its safety and health effects have been studied extensively (*Haruenkit et al., 2007*; *John et al., 2020*). Mangosteen is non-toxic, non-teratogenic, and non-mutagenic. Mangosteen has no acute or subacute toxicity, as confirmed in a rat model. Further, researchers have determined 5,000 mg/kg as the maximum single oral dose of

**Table 2** The preventive and therapeutic effects of mangosteen in various diseases.

| Function | Study Design | Subject | Outcome | Conclusion | Ref. |
|---|---|---|---|---|---|
| Antioxidant property | *In vitro* | GMR, hexane,ethyl acetate, butanol,water fraction | The SOD level in GMR water fraction and TAS level in GMR ethyl acetate fraction were the highest | High DPPH capture activity | *Tjahjani et al. (2014)* |
| | *In vivo* | GMR | Reduce the redox imbalance and toxicity of liver tissue caused by long-term γ -ray irradiation, and improve the liver function, MDA content, antioxidant enzyme activity and NO level of damaged liver | Protective effect from irradiation-induced injury | *Hassan et al. (2021)* |
| | *In vitro* | MPW | Reduce lipid peroxidation and eliminated DPPH free radical,and inhibit the activities of β-secretase and acetylcholinesterase | Antioxidant and neuroprotective effects | *Oh et al. (2021)* |
| | *In vivo* | MCD | Increase the activities of glutathione peroxidase and catalase, reduce the oxidative stress in muscles, and increase the clearance rate of lactic acid | Reduce muscle fatigue after exercise | *Chang et al. (2020)* |
| Anti-inflammatory properties | *In vivo* | GME | Decrease mRNA level, restored the expression of these genes to normal level and down-regulated the expression of pro-inflammatory cytokines | Antibacterial, anti-inflammatory, and wound healing effects | *Tatiya-aphiradee, Chatuphonprasert & Jarukamjorn (2019)* |
| | *In vivo* | GMR | Reduce inflammatory markers (TNF- α, IL-6 and CRP) and down-regulate transcription factors NF-κB/TGF-B1 in liver tissue after long-term γ-ray irradiation | Protective effect from irradiation-induced injury | *Hassan et al. (2021)* |
| | In clinical | MAEC | Reduce crevicular IL-6, increase the salivary MMP-9, and reduce gingival inflammation | A potential to reduce gingival inflammation in the patients with gingivitis and incipient periodontitis | *Park et al. (2021)* |
| Function | | Study Design | Subject | Outcome | Conclusion | Ref. |
|---|---|---|---|---|---|---|
| Antitumor property | Anti-breast tumor | *In vivo* | α-MG | Trigger PARP cleavage and induce apoptosis through PI3K/AKT signaling pathway targeting RXR α, and inhibits the migration and invasion of breast cancer cells | Inhibition effects of invasion and metastasis of MDA-MB-231 cells | *Zhu et al. (2021)* |
| | | | α-MG | Reduce the production of cancerous compounds | Inhibit the proliferation and apoptosis of multiple breast cancer cells | *Herdiana et al. (2021)* |
| | | *In vitro* | α-MG | Inhibit the expression and intracellular activity of FAS, decrease the intracellular fatty acid accumulation, reduce cell viability, induce apoptosis of human breast cancer cells, increase the level of PARP cleavage products, and weaken the balance between anti-apoptosis and pro-apoptosis proteins of Bcl-2 family | Induce breast cancer cell apoptosis by inhibiting FAS | *Li, Tian & Ma (2014)* |
| | Anti-digestive tumor | In vitro+in vivo | α-MG | Reduce tumor size | Enhance the inhibition of sorafenib on the cell proliferation in HCC cell line | *Wang et al. (2021)* |
| | | *In vitro* | Carnation flavone E | Induce a significant cell cycle stagnation in G0/G1 phase, induce apoptosis and necrosis in colorectal adenocarcinoma and human hepatocellular carcinoma cells | Anti-proliferation/cytotoxicity, induce cell killing effect | *Mohamed et al. (2017)* |
| | | | Mangosteen IV | Induce necrosis and apoptosis in HCT116 cells | Cytotoxic property | |
| | | | α-MG | Induce apoptosis and necrosis of HepG2 cells, and moderate necrosis of HCT116 cells | | |
| | | In clinical+ *In vivo* + *In vitro* | γ-MG | Inhibite GSK3 β-related signal pathway, and increase the level of miR-26b-5p | Overcome the 5- fluorouracil resistance induced by CAFs and the production of CSCs | *Wu et al. (2022)* |

Bi et al. (2023), *PeerJ*, DOI 10.7717/peerj.15329

| Function | | Study Design | Subject | Outcome | Conclusion | Ref. |
|---|---|---|---|---|---|---|
| Diabetes prevention and treatment | | *In vivo* | γ-MG | Inhibit α-amylase/ α-glucosidase through insulin sensitization, promote glucose intake and reduce sugar digestion | Anti-hyperglycemia activity | *Chen et al. (2021)* |
| | | *In vivo* | Xanthones | Garcinone E was found to be the most effective PTP1B inhibitor | PTP1B-inhibitory activity | *Hu et al. (2021)* |
| | | In clinical | Mangosteen extract | Improve the insulin sensitivity | Have a possible supplementary role in the treatment of obesity, insulin resistance, and inflammation | *Watanabe et al. (2018)* |
| Neurological system therapy | AD | *In vivo* | MX-IV | Reduce the oxidative stress, neuroinflammation and apoptosis via decrease the brain contents of MDA, H2O2, TNF- α and IL-6 and increase the GSH, and through the regulation of PI3K/Akt/GSK-3 β pathway, decrease the content of NADPH oxidase and the activity of cleaved caspase-3, decrease the number of amyloid plaques and the expression of phosphorylated tau, enhance the neuron survival and cognitive ability | Neuroprotective effects | *Abdallah et al. (2021)* |
| | | *In vitro* | α-MG | Inhibit the cell death induced by oxidative stress in neurons by reducing BAX protein, caspase-3/7 activation and increasing anti-apoptotic BCL-2 protein, bind to the active site of SIRT1, and activate SIRT1/3-FOXO3a pathway | Potentially use as a promising anti-oxidative stress therapeutic compound to treat AD. | *Ramage et al. (2004)* |
| | PD | In vitro+in vivo | TA | Reduce apoptotic cell death in primary cortical neurons, attenuate the behavioral dysfunctions and dopaminergic neuron loss | Cytoprotective effect to dopaminergic neurons | *Huang et al. (2020)* |
| | | *In vitro* | α-MG | Inhibit NF-kB and NADPH oxidase | Protactive effect to neurotoxicity. | *Hu et al. (2016)* |
| | Depression | *In vivo* | GME | Reverse hippocampal lipid peroxidation | Antidepressant-like effects. | *Oberholzer et al. (2018)* |

Bi et al. (2023), *PeerJ*, DOI 10.7717/peerj.15329

| Function | | Study Design | Subject | Outcome | Conclusion | Ref. |
|---|---|---|---|---|---|---|
| Antiparasitic properties | Antimalarial properties | *In vivo* | γ-MG | InteractE with QR-2 includes hydrogen bond and aromatic-aromatic interaction | Inhibit QR-2 | *Liang et al. (2020)* |
| | | *In vitro* | GME | IC50 range from 0.41 to >100 μg/mL | Antimalarial activity and synergistic antimalarial activity with artemisinin | *Tjahjani (2017)* |
| | | *In vitro* | α-MG | Have more active against the resistant Plasmodium FCR3 strain than δ-mangostin | Antimalarial effect | *Upegui et al. (2015)* |
| | Anti-Acanthamoeba properties | *In vitro* | Mangosteen extract and α-MG | Inhibite the growth of Amoeba | Anti-Acanthamoeba activity | *Sangkanu et al. (2021)* |
| | | *In vitro* | Extract and pure compounds from mangosteen | Remove A. triangularis trophozoites within 24 h | Anti-Acanthamoeba activity | *Sangkanu et al. (2022)* |
| Other functions | | *In vivo* | α-MG | Alleviate behavioral alterations, increase the levels of all inflammatory markers, Bax, and caspase-3 | Therapeutic effect on neuropathic pain caused by chronic compression injury | *Ghasemzadeh Rahbardar, Razavi & Hosseinzadeh (2020)* |
| | | *In vivo* | α-MG | Ameliorate doxorubicin cardiotoxicity in human chemotherapy without reduction in its anticancer effect | Protective effects on doxorubicin-induced cardiotoxicity | *Eisvand et al. (2022)* |
| | | In vitro+in vivo | α-MG | Inhibit the formation of osteoclasts | May be a potential choice for the treatment of osteoclast-related diseases | *Zhang et al. (2022)* |

**Notes.**

AD, Alzheimer's disease; CAFs, Cancer-related fibroblasts; CRP, C-reactive protein; CSCs, Cancer stem cell-like cells; DPPH, 1,1-diphenyl-2-picrylhydrazyl; FAS, Fatty acid synthase; FCR3, Falciparum chloroquine-resistant; FOXO3a, Forkhead box protein O3a; GME, Garcinia mangostana Linn. Pericarp extract; GMR, Garcinia mangostana L rind; GSH, Glutathione; GSK3 $\beta$, Glycogen synthase kinase 3 $\beta$; HCC, Hepatocellular carcinoma; $H_2O_2$, Hydrogen peroxide; IC50, Inhibitory concentration; IL-6, Interleukin-6; MAEC, Mangosteen and propolis extracts; MCD, Mangosteen concentrate drink; MDA, Malonaldehyde; MMP-9, Matrix metalloproteinase-9; MPW, Mangosteen peel powder; MX-IV, Mangostanone IV; NADPH, Nicotinamide adenine dinucleotide phosphate; NF- $\kappa$B, Nuclear factor $\kappa$-B; NO, Nitric oxide; PARP, Poly ADP-ribose polymerase; PD, Parkinson's disease; PI3K, Phosphatidylinositol 3 kinase; PKB, Protein kinase B; PTP1B, Protein tyrosine phosphatase 1B; QR-2, Quinone reductase 2; RXR $\alpha$, Retinoid × receptor $\alpha$; SIRT1, Sirtuin 1; SOD, Suberoxide dismutase; TA, Tovophyllin A; TAS, Total antioxidant; TGF-B1, Transforming growth factor-B1; TNF- $\alpha$, Tumor necrosis factor- $\alpha$.

mangosteen extract, while that of $\alpha$-MG is 2,000 mg/kg (*Bunyong et al., 2014*). Towatana et al. found that the direct bilirubin dose-variation increased in rats treated with a mangosteen extract administered *via* an oral gavage without inducing pathological changes. However, several studies demonstrated that increasing doses of mangosteen could induce mortality. The half-lethal dose of a mangosteen extract and $\alpha$-MG administered *via* intraperitoneal injections was 231 mg/kg and 150 mg/kg, respectively (*Choi et al., 2014*). Another study showed that $\alpha$-MG is potentially teratogenic to zebrafish embryos, as an increased mortality rate, cardiac dysfunction, disruption of embryonic ROS balance, and erythropoiesis were observed (*Kittipaspallop et al., 2018*). In addition, $\alpha$-MG-induced intestinal dysbiosis with colitis was also found by some researchers (*Gutierrez-Orozco et al., 2014*).

Although the safety of mangosteen has been demonstrated in animals, it has not yet been proven in humans. A clinical trial of a mangosteen-based formula resulted in no negative effects on the human liver (*Xie et al., 2015*). *Suthammarak et al. (2016)* reported that the oral administration of a polar fraction of a mangosteen pericarp extract had no serious negative consequences. In the treatment of neurological diseases, mangosteen showed no side effects and toxicity. There were statistically important differences in total cholesterol and low-density lipoprotein, but the mean differences from baseline were not significant (*Muangpaisan et al., 2022*; *Turner et al., 2021*).

## CONCLUSION

Mangosteen, a functional food belonging to the *Garcinaceae* family, has numerous pharmacological effects, including anti-oxidative, anti-inflammatory, and anti-cancer properties as well as protective effects against neurological diseases, diabetes, and cardiovascular disease. This review comprehensively summarized the traditional applications, botanical and chemical compositions, and pharmacological effects and applications of mangosteen. Further, it revealed the mechanism of mangosteen's actions in health and disease, providing a theoretical basis for its future clinical use, for example, the development of mangosteen series of food or health products used to prevent and treat various diseases. Moreover, unexplored areas still exist, and the next phase of pharmacological application trials of mangosteen should be a main focus of research in the near future.

## ACKNOWLEDGEMENTS

We thank LetPub for linguistic assistance during the preparation of this manuscript.

### Funding

This work was funded by Zhejiang Provincial First Courses of Medical Biochemistry (No. 20220646) and Yue Di Element General Curriculum Project of Shaoxing Higher Education Institutions (No. 20225306). The funders had no role in study design, data collection and analysis, decision to publish, or preparation of the manuscript.

## Grant Disclosures

The following grant information was disclosed by the authors:

Zhejiang Provincial First Courses of Medical Biochemistry: 20220646.

Yue Di Element General Curriculum Project of Shaoxing Higher Education Institutions: 20225306.

## Competing Interests

The authors declare there are no competing interests.

## Author Contributions

- Chenchen Bi conceived and designed the experiments, performed the experiments, analyzed the data, prepared figures and/or tables, authored or reviewed drafts of the article, and approved the final draft.
- Hang Xu conceived and designed the experiments, performed the experiments, analyzed the data, authored or reviewed drafts of the article, and approved the final draft.
- Jingru Yu performed the experiments, analyzed the data, prepared figures and/or tables, and approved the final draft.
- Zhinan Ding performed the experiments, analyzed the data, authored or reviewed drafts of the article, and approved the final draft.
- Zheng Liu performed the experiments, analyzed the data, authored or reviewed drafts of the article, and approved the final draft.

## Data Availability

This is a literature review.

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
