# Peer review of "Botanical characteristics, chemical components, biological activity, and potential applications of mangosteen"

_PeerJ, doi:10.7717/peerj.15329_

## Round 0.1 · original submission · Major Revisions

Dear Authors

This manuscript was reviewed by the three reviewers and they find merit. It is requested to the authors to revise the manuscript and resubmit it for consideration

Reviewer 1 ·

Basic reporting

Reviewer comments:
Dear Editor,
Thank you very much for giving this opportunity to review the manuscript entitled “Botanical characteristics, chemical components, biological activity, and potential applications of mangosteen” for this reputed Journal. The review article was well designed and touches almost every aspect of the topics mentioned in the Title. But the manuscript is still not ready for publication. The decision over the manuscript is “Major Revision”. All the required corrections are highlighted inside the manuscript in yellow color with attached comment boxes. Authors are asked to go through all of them and correct them. All the comments are enlisted below also-
Comments:
Title: Line number 1-3:
Reconstruct the title in more attractive and scientifically sound way.
Write the scientific name with the identifier name instead of just writing as Mangosteen.
Abstract:
Line number 23:
It is obvious that this plant has not been used all over the world, right?? It should must be used by some particular community or by a particular group of people of a specific place. Author must be very precise while writing such kind of statements.
Line number 24-25:
Instead of writing "and" author could have write "with", which would have improved the importance of the phytochemicals present in the plant sample.
This statement is already written in the very beginning of this abstract. Avoid repetition of sentences with same meaning in different styles.
Line number 26:
Before writing this statement author need to write about the sources or the methodologies (such as search engines, books, literatures etc.) used for proper review of the topic.
Line number 27:
Instead of writing the name mangosteen, author should first the write the scientific name of the species and after that author can use the word mangosteen.
Line number 30-31:
Rewrite the sentence in a more attractive way.
Introduction:
Line number 33: Did the author not know that scientific name should must always be written in italics?? Do not repeat such kind of mistake in near future.
Line number 35-36: At exactly what altitude (the range ---to--- m) most of the population of this species found?? Need to mention it in this statement.
Line number 39-40: This statement seems to be incomplete. After this statement author need to add one more sentence supporting the above-mentioned statement.
Line number 42: "attracted attention recently"- recently means from which year?? Need to cite reference from that particular year to till date. Which is the country of origin?? Name the country with appropriate supporting reference.
Line number 50-54: The gaps of the study is not well established in the introduction section. Author needs to critically review all the published works and need to address the missing loop holes as point of discussion in this review article. In short author failed to establish the novelty of this review literature in the introduction section.
Survey Methodology:
No book chapters, short communications, internet available data such as newsletter etc., were used or what?? Write about all the materials used while conducting this review in this section of the manuscript.
Traditional application:
Line number 61, 62, 63: Write the names of the countries. Ayurvedic system belongs to which community of people?? Cite references in support of this statement. What did author mean by "wide α mangosteen,???
Botany:
Line number 81,82: Why scientific names are not written in italics?? This is okay but how exactly the name came into existence?? Author need to explain these things as well. Additionally, how the name "Mangosteen" came into existence it should also be mentioned with appropriate references.
Line number 86, 87, 89: Why this reference was cited in this way??
Cite references in accordance to the Journal format only. Author needs to see the authors guideline section or can check a recently published review article of this Journal.
Mangosteen: Mangosteen plants.
Line number 92: This is not the proper way of writing references for this Journal.
Phytochemistry:
Polysaccharides: Line number 103-104: As far as immunomodulatory concern.. there must be some well-known examples of polysaccharides with such properties. Author needs to mention few examples in this statement with appropriate reference.
Line number 113: Where is the year??
Line number 123-127: Need to be written in authors own words with proper explanations.
Line number 128-129: Author can add some examples of polysaccharides used in the field of scientific research not necessarily from this particular plant only.
Line number 130: potential in the cosmetic....
Line number 133: Thats okay.. So what else could be done in near future?? What are the authors own input in this regard? These are the things that need to be included in the revised version of the manuscript.
Xanthone:
Line number 152: In vivo in particular which field?? It is not possible that it will show all kinds of activities, right?? Try to be specific as much as possible. Need to cite appropriate references in support as well.
Line number 182: Jianget al., 2010 should be in italics as Jiang et al., 2010. Spacing issue, formatting issue etc., are there in all the references. Author needs to revise the cited references in Journal format.
Pharmacology:
Line number 196-197: Give some examples of antioxidants.
Line number 216-218: There are numerous numbers of antioxidant drugs are available in the market then why someone will go for natural antioxidants?? Need clarifications in this aspect.
Anti-inflammatory properties:
Line number 220: What kind of stimulation will initiate inflammation??
Line number 231: Mangosteen, what??? Which part, which extract??
Line number 269-270: As mentioned by the author that it has anticancer properties and from literature survey also, we can see that there are numerous reports available in public domain regarding anticancer potential of lots of plant extracts. But what is the reason that in modern world there are still not a single particular drug available for treatment of cancer?? What is the probable reasons author can discuss this in this section as well.
Line number 279-281: Author mentioned only one type of diabetes what about the other one?? Is there any research conducted on that also??
Line number 305: it should be investigation not investigated.
Conclusion:
The untouched or unexplored areas need to be mentioned in the conclusion section. Where next level of trials for pharmacological application can be conducted which should be main focus point for near future need be clearly mentioned in the conclusion section as well.
References:
All the references must be in accordance to the Journal format only. Author needs to check the authors guideline section or can see a recently published review article from the Journal for proper formatting of the References.
Table 2: All the in vitro and in vivo words must be presented in italics form only throughout the whole manuscript.
Additionally, the language of the manuscript must need to be improved and it should be free of any grammatical or typical errors.

Experimental design

No comments

Validity of the findings

No comments

Additional comments

No comments

Annotated reviews are not available for download in order to protect the identity of reviewers who chose to remain anonymous.

·

Basic reporting

Based on my review, the authors did a comprehensive review and meet the high quality for the journal.

Experimental design

No comment

Validity of the findings

No comment

Additional comments

- The authors need to perform extensive English editing by the fluent speaker.
- Some figures need to be improved.

Reviewer 3 ·

Basic reporting

This review is very comprehensive and expertly written, and it meets the standards of PeerJ. This article provides a good introduction to the topic.

Experimental design

The methodology of the survey is consistent, and adequate references are cited

Validity of the findings

The conclusion is inadequate, Future research directions and gaps are not discussed

Additional comments

Authors should discuss the gaps and future directions

---

## Round 0.2 · Minor Revisions

Resubmit the manuscript as per comments of the reviewers for consideration.

Reviewer 1 ·

Basic reporting

Reviewer comments:
The manuscript “#81111” entitled “Characteristics, Active Components, Pharmacological Effects, and Potential Applications of Garcinia mangostana L.: A Review” is well revised but there are still some corrections required before being considered for publication. The decision over the manuscript is “Minor Revision”. All the required corrections are enlisted below.
Comments:
1. Author has removed the name of Ayurvedic system. Why?? Need to re-add and explain it.
2. How exactly the name came into existence not explained in the revised version of the manuscript, why??
3. Some of the references Example., Zhang et al.,…. Year is still missing inside the manuscript.
4. Author still needs to add some examples of Cancer medicines available in the present-day pharmaceutical industries and need to explain and compare them with the potential natural candidates with anti-cancer properties.
5. Overall, literature survey and in-depth scientific discussion is still required in this aspect.
6. “Gutierrez-Orozco, F., Thomas-Ahner, J. M., Berman-Booty, L. D., Galley, J. D., Chitchumroonchokchai, C., Mace, T., ... Failla, M. L. (2014). Dietary α-mangostin, a xanthone from mangosteen fruit, exacerbates experimental colitis and promotes dysbiosis in mice. Mol Nutr Food Res, 58(6), 1226-1238. 10.1002/mnfr.201300771.”
-Why there are “….” Written after Mace, T., ….???

Experimental design

No comment

Validity of the findings

No comment

Additional comments

No comment

·

Basic reporting

Firstly, this paper is clear and well-organized. However, I may require some comments on the following issues.
Title: The title is easy to follow and has no mistakes.
Abstract: This section was well-written and easy to understand. In addition, the objective of the study and the state of the art of the study is clear. Furthermore, the keywords should be represented the study.
Introduction: Based on my review, I recommend that the authors should add various improvements, such as:
- Hypothesis and objectives.
- Add a statement or sentence about whether there are any similar studies that have been done before.
- Add the essential issues.
Conclusion: In this part of the manuscript, the authors have successfully established the conclusion of this research. However, it might be useful if the authors mention the suggestion for forthcoming research.

Experimental design

Clear

Validity of the findings

Adequate.

Additional comments

- English editing is mandatory for this manuscript.
- References: The core references are acceptable, but I think the authors should add the latest references in this manuscript. Please mentioned many high-quality related papers, such as:
> https://doi.org/10.5958/0974-360X.2020.00182.1

---

## Round 0.3 · accepted · Accept

All the comments have been resolved properly.